# Tailoring Digital Tools to Address the Radiation and Health Information Needs of Returnees after a Nuclear Accident

**DOI:** 10.3390/ijerph182312704

**Published:** 2021-12-02

**Authors:** Takashi Ohba, Aya Goto, Yui Yumiya, Michio Murakami, Hironori Nakano, Kaori Honda, Kenneth E. Nollet, Thierry Schneider, Koichi Tanigawa

**Affiliations:** 1Department of Radiation Health Management, School of Medicine, Fukushima Medical University, Fukushima 960-1295, Japan; 2Center for Integrated Science and Humanities, Fukushima Medical University, Fukushima 960-1295, Japan; agoto@fmu.ac.jp (A.G.); y-yumiya@fmu.ac.jp (Y.Y.); kaori-h@fmu.ac.jp (K.H.); 3Department of Public Health and Health Policy, Graduate School of Biomedical and Health Sciences, Hiroshima University, Hiroshima 734-0037, Japan; 4Department of Health Risk Communication, Fukushima Medical University School of Medicine, Fukushima 960-1295, Japan; michio@fmu.ac.jp; 5Radiation Medical Science Center for the Fukushima Health Management Survey, Fukushima Medical University, Fukushima 960-1295, Japan; h-nakano@fmu.ac.jp; 6Department of Epidemiology, Fukushima Medical University School of Medicine, Fukushima 960-1295, Japan; 7Department of Blood Transfusion and Transplantation Immunology, Fukushima Medical University School of Medicine, Fukushima 960-1295, Japan; nollet@fmu.ac.jp; 8Nuclear Protection Evaluation Center (CEPN), 92260 Fontenay-aux-Roses, France; thierry.schneider@cepn.asso.fr; 9Futaba Medical Center, Tomioka Town, Fukushima 979-1151, Japan; tanigawa@futaba-med.jp

**Keywords:** application tool, eHealth, Fukushima nuclear accident, health promotion, KAP survey, radiation protection, SHAMISEN-SINGS project

## Abstract

Digital tools are increasingly used for health promotion, but their utility during recovery from a nuclear disaster has yet to be established. This study analysed differences in knowledge, attitude, and practice (KAP) toward digital tools for radiation protection and health promotion, and preferences for specific application functions, among cohorts living within and outside areas affected by the Fukushima Daiichi nuclear power station (FDNPS) accident. A needs assessment was conducted by internet survey, and responses from those affected (N = 86) and not affected (N = 253) were compared and quantified by an adjusted odds ratio (aOR), using logistic regression analyses. KAP toward the radiation-related application in the affected group had an aOR of 1.95 (95% confidence interval (CI) = 1.12–3.38) for knowledge, and 5.71 (CI = 2.55–12.8) for practice. Conversely, toward the health-related application, the aOR of the affected group was 0.50 (CI = 0.29–0.86). The preference in the affected group was significantly lower for two application functions related to radiation measurement and two health-related functions (one about the effects of radiation in general and another about personal health advice in general): aOR range 0.43–0.50. Development of specific applications incorporating the findings from this survey was intended to foster a locally appropriate eHealth environment during recovery from the FDNPS accident.

## 1. Introduction

The recovery phase after a nuclear accident is complex, involving not only radiation protection, but also social and environmental considerations, for people returning to the affected area [1]. Those who evacuated after the 2011 Fukushima Daiichi nuclear power station (FDNPS) accident continue to suffer from negative physical and mental health consequences [2,3]. Thus, it remains immensely challenging to foster a favourable social environment for returning populations in the affected area, not only for them to implement radiation protection measures, but also to improve their physical and mental health and the quality of their daily life [4].

Digital devices and application tools are now widely available, with interactivity fostering user engagement. The European Union’s Nuclear Emergency Situations—Improvement of Medical and Health Surveillance—Stakeholder Involvement in Generating Science (SHAMISEN-SINGS) project developed recommendations for using digital devices and application tools for the timely sharing of radiation dose and personal health information among local stakeholders and affected populations in the early and long-term recovery from a nuclear accident [5]. The World Health Assembly Resolution on Digital Health recognized the importance of “eHealth” for achieving universal health coverage and recommended that government health ministries “assess their use of digital technologies for health… and to prioritize, as appropriate, the development, evaluation, implementation, scale-up and greater use of digital technologies...” [6]. Eysenbach, an editor of the Journal of Medical Internet Research wrote that “eHealth is an emerging field in the intersection of medical informatics, public health and business, referring to health services and information delivered or enhanced through the Internet and related technologies.” [7]. For the eHealth movement to truly benefit support providers and returnees after a nuclear accident, digital information needs of returnees should be carefully assessed as a first step.

In the case of the FDNPS accident, returnees to the municipalities near the FDNPS have received support for radiation protection and health promotion through multiple channels and by various professionals, including radiation specialists and health care workers [8,9,10]. It has been reported that some returnees learned to make their own radiation protection decisions through community self-support activities and dialogues with experts [11,12]. As digital information tools are introduced, new communication modes are expected to link residents with various available resources more effectively and to streamline communications among residents and support providers. Soon after the Fukushima nuclear accident, several municipalities distributed digital tablets and other internet-related tools. However, the Mitsubishi Research Institute’s survey among municipal office personnel and volunteers reported that the disaster-affected elderly preferred printed information materials and, for digital tools, support (e.g., easy-to-use, large-screen devices, and training sessions) [13].

This study aimed to compare knowledge, attitude, and practice (KAP) toward digital tools for radiation protection and health promotion between people living within and outside areas affected by the FDNPS accident, and to ascertain what types of information they would want from these tools. Here, KAP analysis follows the WHO definition as “a representative study of a specific population to collect information on what is known, believed, and practiced in relation to a particular topic” [14]. The KAP survey provides information on the needs, problems, and barriers associated with the development of effective public health interventions tailored to local situations [14]. More recently, the KAP methodology has been used to investigate health behaviours and risk behaviours related to coronavirus infectious disease 2019 (COVID-19), and recommendations for education and policy are being developed to bridge the gap between knowledge and practice [15]. Results obtained in the present study will guide the actual development and piloting of a mobile application for radiation protection and health promotion for returnees in Fukushima [16].

## 2. Materials and Methods

### 2.1. SHAMISEN-SINGS Project

The SHAMISEN-SINGS project’s overall goals have been to strengthen public participation in stakeholder and expert decision-making, to share radiation exposure data (doses and dose rates), and to improve the health and well-being of those affected by a nuclear accident using their recommendations (Appendix A) [5]. Since application tools that are adapted to a nuclear accident have little or no precedent, this project implemented a needs survey of the general public. Our study used a Japanese version of the SHAMISEN-SINGS questionnaire survey, funded by Japan’s Ministry of the Environment and conducted in collaboration with the SHAMISEN-SINGS project [16].

### 2.2. Survey Protocol for Two General Public Subject Groups

The protocol used in this internet survey aligned with the SHAMISEN-SINGS project in terms of subject allocation and questionnaire content [5]. The subjects of this study were from two groups, as shown in Table 1. The group affected by the FDNPS accident included residents of the Hamadori and Nakadori areas of Fukushima Prefecture (Figure 1). Hamadori is the coastal area where the FDNPS is located; inland Nakadori also received fallout from the accident. As a control, the non-affected group consisted of residents living both within and outside 30 km radii of other NPSs elsewhere in Japan (a total of 116 municipalities). The study adapted the SHAMISEN-SINGS protocol as follows. Due to the relatively few areas that have ever experienced NPP accidents in the world, the SHAMISEN-SINGS protocol compared populations living within and outside 30 km radii of NPSs with populations living in other areas. As a part of Japan was affected by the FDNPS accident, we merged residents living within and outside 30 km radii of NPSs with those living in other parts of Japan for this analysis. Each Japanese NPS and its 30 km perimeter are represented as grey dots surrounded by dashed lines in Figure 1. The intended number of subjects in each group was 75 and 210, respectively, with 25 subjects in each of three age groups (20 s, 30–50 s, and 60 or older), and 20 subjects in each of three stakeholder groups (government, medical personnel, and teaching staff) (Table 1). The recruitment, administration, and collection of the questionnaires for this study were performed by INTAGE Research Inc. (Tokyo, Japan) using their internet survey system. The survey took place in 2020, between 31 January and 4 February.

INTAGE Research Inc. is the largest Japanese research company of its kind, engaging over 3 million survey participants annually. This company can collate responses by occupation, age range, and living area for the research target group. In addition, INTAGE implements two quality controls: (1) quality control of survey participants (e.g., locating survey participants by mail-outs and removing inappropriate participants) and (2) quality control of the content of response (e.g., excluding answers with too short a response time or multiple responses from the same IP address). Survey participants can receive rewards (e.g., cash, internet points, gift certificates) for their responses. The specific characteristics of this company’s online survey have already been reported [17].

### 2.3. Survey Protocol for Questionnaire

The questionnaire for this study was prepared by rendering the English version of the SHAMISEN-SINGS questionnaire into Japanese. Questionnaires solicited basic characteristics (gender, age range, education level, occupation, living conditions), knowledge of radiation (with four options: limited, average, professional, or none), and concern about the risks of NPSs (options: yes, sometimes, or no). The KAP questions on radiation-related devices or applications were as follows: “Are you aware of existing mobile applications or personal devices that allow you to perform your own radiation dose measurements?” (K—knowledge), “Would you be interested in using mobile applications that allow you to measure radiation?” (A—attitudes), and “Have you ever used any of these mobile applications or devices to measure radiation dose?” (P—practice). Here, radiation dose measurements were defined as dose-related information, including measurements of ambient dose rates and overall dose assessment; we also asked about the subject’s knowledge and attitudes, including interest and concerns. Those for health-related applications were as follows: “Are you aware of existing mobile applications that allow you to monitor your health status?” (K—knowledge), “Would you be interested in using an application or device that allows you to measure/monitor your health status and well-being during and after a radiation accident?” (A—attitudes), and “Have you used any of these mobile applications that allow you to measure/monitor your health status?” (P—practice). Here, the definition of health was from the WHO constitution, “Health is a state of complete physical, mental and social well-being and not merely the absence of disease or infirmity” [18]. We aspire to Goal 3 of the WHO’s Sustainable Development Goals (SDGs), “Ensure healthy lives and promote well-being for all at all ages,” in this study [16,19]. Therefore, the definition of well-being in this study comprehensively included the above concepts.

In addition, if an answer to the attitude item was “sometimes” or “yes,” an additional question was asked with multiple answers as to what application functions would be of interest. Answer options for the radiation-related devices or applications were as follows: Radi (1) Measure environmental radiation levels; Radi (2) Measure radiation levels in food and other consumable products; Radi (3) Provide real-time information on the current situation (official channels only); Radi (4) Provide real-time information on the current situation (non-government channels); Radi (5) Provide general information on the effects of radiation on health and protection measures; Radi (6) Provide specific instructions related to my personal situation and status; Radi (7) Provide some degree of interactivity (questions and answers, live chat); and Radi (8) Others. Multiple answer items for health-related applications were as follows: Health (1) Measure health parameters (e.g., weight, blood pressure, and blood sugar); Health (2) Collect physical activity data (e.g., number of steps); Health (3) Collect information on mental health and/or well-being; Health (4) Provide general information on the effects of radiation on health and on protection measures; Health (5) Provide specific advice and instructions related to your personal situation and health status and/or well-being; Health (6) Provide some degree of interactivity (questions and answers, live chat); and Health (7) Others.

### 2.4. Data Analysis

Univariate analysis of differences in KAP between affected and non-affected groups used the chi-square test or Fisher’s exact test. The same analysis was repeated after stratifying by age (20–50 s and 60 or older), since older age groups tended not to be familiar with digital information. The same analysis strategy and age stratification were used to assess preferences for radiation- and health-related application functions. Multivariable analyses for the KAP survey and preferences for radiation- and health-related application functions were performed using a logistic regression model, adjusting for basic characteristics (gender, age range, education level), knowledge of radiation with four choices (none, limited, average, and professional), and concern about the risks of NPSs. Our analysis of the knowledge of radiation with four answer options was carried out by setting dummy variables for the multivariable analysis. The adjustment items for multivariable analyses excluded occupation, which correlated with education level, and living conditions, which did not show significance in the univariate analysis.

Analyses were performed with IBM SPSS Statistics, version 25.0.0.2 (IBM Corp., Armonk, NY, USA). All significance levels were set at 5%.

### 2.5. Ethical Considerations

Research objectives and procedures were included in the internet survey form. We considered participants to have given their consent by completing the questionnaire. This study was approved by the Ethics Committee of Fukushima Medical University (approval number: 2019-133).

## 3. Results

### 3.1. Characteristic Information for the Subjects

We received 339 answers, which was 119% of the target 285. The number of respondents in the affected group was 86 (115% of the target 75), and that in the non-affected group was 253 (120% of the target 210). Table 2 shows the basic characteristics of the two groups. Significant differences were found between the two groups in terms of education level and occupation. In addition, the percentage of those concerned about the risks of NPSs (question: “Are you personally concerned about potential dangers and risks related to living close to NPSs?”) was significantly higher in the affected group. Among those who answered “yes/sometimes” (n = 274) to this question, the reasons were as follows: “potential effects on you or your family’s health even in the absence of an accident” for n = 133 (affected group: 40, non-affected group: 93), “possible occurrence of a nuclear accident” for n = 194 (affected group: 49, non-affected group: 145), and “others” for n = 9 (affected group: 1, non-affected group: 8).

### 3.2. KAP of the Two-Age Groups

Table 3 shows differences in KAP for radiation-related devices and applications and health-related applications between the two groups. Regarding radiation-related devices and applications, proportions of positive responses to knowledge and practice in the affected group were significantly higher than those of the non-affected group. Stratified by age, similar trends emerged for knowledge with a borderline significance for practice in the 20 s–50 s age group and practice in the 60 or older group. Regarding health-related applications, the overall proportion of positive responses to attitudes (having an interest) in the affected group was significantly lower than in the non-affected group (*p* = 0.02), with borderline (*p* = 0.09) significance in each of the smaller subgroups defined by age range.

Table 4 shows the results of logistic regression analyses for the KAP of radiation-related devices and applications and that of health-related applications, with the residential group as an independent variable. With regard to the radiation-related tools, the affected group showed a significantly higher odds ratio even after adjusting for basic attributes; the adjusted odds ratios (aOR) were 1.95 (95% confidence interval (CI) = 1.12–3.38) for knowledge and 5.71 (CI = 2.55–12.8) for practice. Conversely, the aOR of the affected group for attitudes toward health-related applications was 0.50 (CI = 0.29–0.86).

### 3.3. Differences in Preferred Application Functions between the Two Residential Groups

Differences in the choice of application functions between the two residential groups are shown in Figure 2 for radiation-related devices and applications and in Figure 3 for health-related applications. Regarding radiation-related devices and applications (Figure 2), the proportion of those who selected “Radi 1”and “Radi 3” was significantly lower in the affected group than in the non-affected group. This statistical significance was observed only for the 20 s–50 s age range, but the trend was the same in the older age group. Regarding the health-related applications (Figure 3), the proportion of those who selected “Health 4” and “Health 5” was significantly lower in the affected group than in the non-affected group. Statistical significance diminished when stratified by age, but the trend was the same in both age groups.

Table 5 shows the logistic regression analyses for the relationship between preference of application functions and residential group. The affected group showed a significantly lower odds ratio, even after adjusting for basic attributes: aORs of 0.50 (CI = 0.26–0.95) for “Radi 1”, 0.43 (CI = 0.23–0.81) for “Radi 3”, 0.48 (CI = 0.24–0.96) for “Health 4”, and 0.50 (CI = 0.25–0.99) for “Health 5”.

## 4. Discussion

This study compared the needs for radiation-related and health-related application functions between groups affected and not affected by the FDNPS accident, about a decade after the accident occurred. The results informed the design of an application for radiation protection and health promotion that can address people’s needs during the recovery phase of a nuclear accident.

### 4.1. Needs for Radiation-Related Devices and Applications

Results of the KAP analyses showed that knowledge and practice of the affected group were significantly higher than those of the non-affected group with regard to radiation-related devices and applications (Table 3 and Table 4). This was particularly noticeable for practice in both younger and older age groups (Table 3). This result confirms that, since the FDNPS accident, the affected residents felt the need to know about and measure radioactive materials released from the FDNPS for self-protection. Information on radiation protection was often disseminated through digital tools, such as the internet, after the FDNPS accident [20,21,22]. In Japan, around 95% of people in their 20 s–50 s have internet access [23]. As for radiation measurements, municipal offices in Fukushima Prefecture distributed personal dosimeters for residents to facilitate understanding of their own environment [8,9,11,12,24]. After the Chernobyl NPS accident, support to the local population was provided by using radiation measurements, but digital communication technology was not yet developed [25]. It is now possible to provide community-based support to the returnees by incorporating new technologies, as in our project.

Interestingly, preferences for a few application functions were lower in the affected group than in the non-affected group, especially in those of younger age (Figure 2 and Table 5). These functions included radiation measurements (“Radi 1”) and provision of official real-time information (“Radi 3”) (Figure 2). We did not ask reasons for respondents’ preferences of application functions, but it is possible that they have lost interest in radiation itself or have ignored information about radiation doses almost a decade after the accident. Information on radiation measurements was readily available in Fukushima Prefecture, including information on changes in air radiation dose rates using monitoring posts, and information on one’s own external exposure using personal dosimeters [24,26]. In addition, residents of municipalities in the affected areas routinely received official real-time information through written materials and online [21,22]. Therefore, this project is not about developing a new radiation monitoring tool but rather about installing the trusted website links in the application to be developed.

Although there was no significant difference between the two residential groups in terms of the tendency to choose the application functions related to “Radi 2” (measure radiation levels in food and other consumable products), some of those who selected the item supported “Radi 1” and “Radi 3”. Concurrent with a decreasing trend in local food avoidance [27], the residents in areas affected by the FDNPS accident may still need to measure foodstuffs to minimise internal exposure [28]. Thus, the radiation-related application functions should include a simple recording function of internal exposure, such as the data of whole-body counting and food measurement. This recording function enables users to look for trends in the data that they enter and facilitate evidence-based communication with health and welfare service providers.

### 4.2. Needs for Health-Related Applications

Attitudes toward the health-related applications were lower in the affected group than in the non-affected group (Table 3 and Table 4). One possible reason was that attention of the affected group skewed toward radiation. We believe that this is largely due to differences in risk perception of radiation versus other health risks. It is indicated that the high level of interest in ICRP Dialogue Seminars and NPO support were due to the high levels of concern about radiation among the affected populations [11,12]. On the other hand, non-affected populations would be less familiar with the risks of radiation and would therefore be more concerned with everyday health risks. Another possible explanation would be that there were various outreach efforts to provide face-to-face health and welfare support in affected areas after the FDNPS accident [10], which led to less demand for health promotion services among returnees.

Similarly, toward the radiation application, preferences of a few application functions were lower in the affected group than in the non-affected group (Figure 3 and Table 5). These functions include providing information on the health effects of radiation (“Health 4”) and providing advice on health status and well-being (“Health 5”) (Figure 3). Populations in the affected area of the FDNPS accident were provided with Q&A booklets such as ‘Provide Basic Information Regarding Health Effects of Radiation’ and ‘Information Booklet for Returnees’ supported by Japan’s Ministry of the Environment [21,22]. As in these cases, the affected group in this study may have tended to be concerned less about radiation than about their usual health risks. There were different vectors of needs in application functions between the affected and non-affected groups.

Preferences in application functions related to physical and mental indicators such as Health 1, Health 2, and Health 3 in Figure 3 were equally high in both residential groups. These functions were not directly related to the health effects of radiation exposure and radiation protection. In the case of the FDNPS accident, the health status of the affected residents has shown signs of improvement, but not yet to pre-accident levels [29]. Active digital information usage for health promotion is recommended in the recovery phase from a nuclear accident, so functions related to this aspect should be included in the application for returnees.

### 4.3. Limitations of This Study

Some limitations of this study are evident. Firstly, this was a survey of application tools for the affected population about 10 years into the recovery period after the FDNPS accident. This makes it difficult to apply study results to the needs of those in earlier phases of a nuclear accident. Since nuclear accidents vary in scale, in events that follow and in evacuation scenarios, it is necessary to tailor application functions to each accident. Secondly, our survey was conducted online, so respondents may have been more accustomed to digital tools than the general public as a whole. Therefore, we are currently conducting field face-to-face interviews, with a cultural anthropologist, to hear detailed feedback from users of our new application. Finally, we found that in the recommendations of the SHAMISEN-SINGS project shown in Appendix A, the following three points have not been considered: (4) apply incentives to promote usage, (6) involve vulnerable populations, and (7) accommodate multiple languages. In our previous reports, we suggested the following functions related to application tools [16,30]: (1) automatic replies with advice (to maintain motivation in the ongoing use of the application tool) and (2) icons (pictograms) so the elderly, immigrants, and any others who struggle with the Japanese language can better understand the application functions. Including these features should be considered as our digital tools are upgraded.

## 5. Conclusions

Our results provided practical strategies for developing an application to be used among people affected by the FDNPS accident (nuclear disaster), including those from former evacuation areas. First, functions for radiation protection can be simplified, but should include items related to both external and internal exposure and links to reliable information sources. Second, health promotion functions need to be expanded. Third, multilingual options and other features, such as pictograms, to accommodate vulnerable groups, such as the elderly, immigrants, and children, should be considered when upgrading the application. In addition, a parenting diary function may be of help to mothers [16]. We believe that developing an application tool that incorporates the insights gained from a needs survey would help nurture an eHealth environment suitable for the community during the recovery phase of the FDNPS accident. Flexibility to allow modifications to digital tools will improve their utility to users as different crises emerge and evolve.

## Figures and Tables

**Figure 1 ijerph-18-12704-f001:**
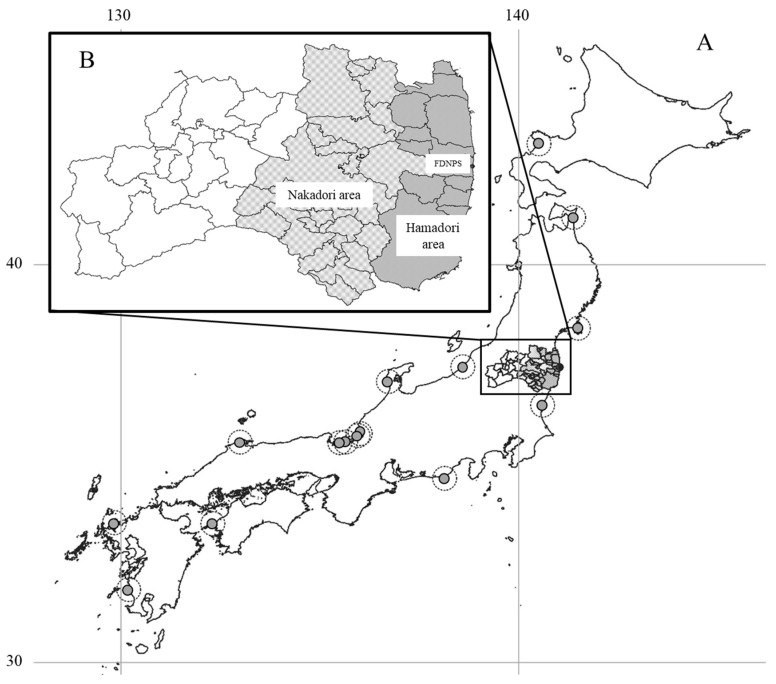
Geographical location of the areas covered by the internet survey: (**A**) Japan, (**B**) Fukushima Prefecture.

**Figure 2 ijerph-18-12704-f002:**
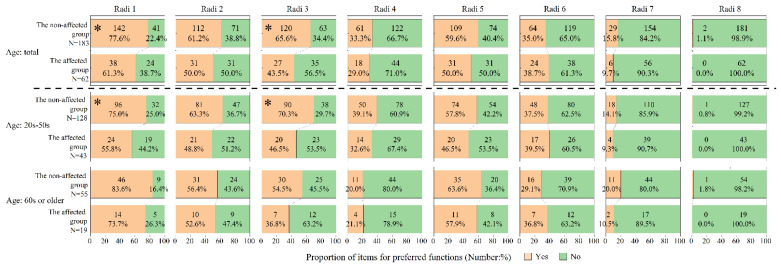
Preferred functions of radiation-related devices and applications ^a^. ^a^ Subjects interested in radiation-related devices and applications were asked to select functions that they preferred. Preferences of application function were as follows: Radi (1) Measure environmental radiation level, Radi (2) Measure radiation level in food and other consumable products, Radi (3) Provide real-time information on the current situation (official channels only), Radi (4) Provide real-time information on the current situation (non-government channels), Radi (5) Provide general information on the effect of radiation on health and protection measures, Radi (6) Provide specific instructions related to my personal situation and status, Radi (7) Provide some degree of interactivity (questions/answers/live chat), and Radi (8) Others. Asterisk (*) indicates *p* < 0.05.

**Figure 3 ijerph-18-12704-f003:**
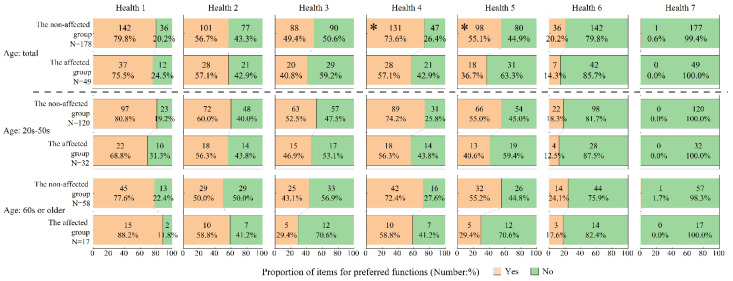
Preferred functions of health-related applications ^a^. ^a^ Subjects interested in health-related applications were asked to select functions that they preferred. Preferences for application functions were as follows: Health (1) Measure health parameters (e.g., weight, blood pressure, and blood sugar), Health (2) Collect physical activity data (e.g., number of steps), Health (3) Collect information on your mental health/well-being through a questionnaire, Health (4) Provide general information on the effect of radiation on health and protection measures, Health (5) Provide specific advice and instructions related to your personal situation and health status and/or well-being, Health (6) Provide some degree of interactivity (questions/answers/live chat), and Health (7) Others. Asterisk (*) indicates *p* < 0.05.

**Table 1 ijerph-18-12704-t001:** Participant allocation and grouping in the internet survey.

Group Category	Subject Group	Occupation	Age Range	Expected Number of Subjects	Expected Total Number
The affected group	Residents affected by the Fukushima accident	-	Young adults (20 s)	25	75
		-	Middle-age adults (30 s, 40 s, 50 s)	25	
		-	Older adults (60 s or older)	25	
The non-affected group	Residents living within 30 km of another NPS ^a^	-	Young adults (20 s)	25	210
		-	Middle-age adults (30 s, 40 s, 50 s)	25	
		-	Older adults (60 s or older)	25	
	Residents living over 30 km from another NPS ^a^	-	Young adults (20 s)	25	
		-	Middle-age adults (30 s, 40 s, 50 s)	25	
		-	Older adults (60 s or older)	25	
	Stakeholders	National and local authorities	-	20	
		Medical workers	-	20	
		School teachers	-	20	

^a^ NPS: Nuclear power station.

**Table 2 ijerph-18-12704-t002:** Individual characteristics of two residential groups, affected and not affected by the FDNPS accident.

	Total N = 339	Numbers by Characteristic (%)		*p* Value (Chi-Square)
		The Affected Group N = 86	The Non-Affected Group N = 253	
Characteristic				
Age				
20 s–50 s	239	59 (68.6)	180 (71.1)	0.66
60 s or older	100	27 (31.4)	73 (28.9)	
Gender				
Female	166	43 (50.0)	123 (48.6)	0.82
Male	173	43 (50.0)	130 (51.4)	
Education				
High school or lower junior college, technical school	182	60 (69.8)	122 (48.2)	<0.01
University, graduate school	157	26 (30.2)	131 (51.8)	
Employment status				
Unemployed	92	31 (36.0)	61 (24.1)	0.03
Employed	247	55 (64.0)	192 (75.9)	
Living status				
Alone	101	30 (34.9)	71 (28.1)	0.23
With a partner and/or children, other	238	56 (65.1)	182 (71.9)	
Knowledge and concerns about radiation				
Presence or absence of knowledge of radiation, based on self-evaluated level				
None	72	14 (16.3)	58 (22.9)	0.61
Limited	86	23 (26.7)	63 (24.9)	
Average	172	47 (54.7)	125 (49.4)	
Professional	9	2 (2.3)	7 (2.8)	
Concerns about potential dangers and risks of living near a nuclear power plant				
No	65	10 (11.6)	55 (21.7)	0.04
Yes/Sometimes ^a^	274	76 (88.4)	198 (78.3)	

^a^ Among those who answered “yes/sometimes” (n = 274), the reasons were as follows: “potential effects on your or your family’s health even in the absence of an accident” for n = 133, “possible occurrence of a nuclear accident” for n = 194, and “others” for n = 9.

**Table 3 ijerph-18-12704-t003:** Differences in KAP between two residential groups: univariate analysis stratified by age. (Number (%)).

Items	Total		*p* Value *	Age (20 s–50 s)		*p* Value *	Age (60 s or Older)		*p* Value *
	The Affected Group N = 86	The Non-Affected Group N = 253		The Affected Group N = 59	The Non-Affected Group N = 180		The Affected Group N = 27	The Non-Affected Group N = 73	
KAP pertaining to radiation-related devices and applications									
Knowledge									
No	51 (59.3)	183 (72.3)	0.02	39 (66.1)	141 (78.3)	0.06	12 (44.4)	42 (57.5)	0.24
Yes	35 (40.7)	70 (27.7)		20 (33.9)	39 (21.7)		15 (55.6)	31 (42.5)	
Attitude									
No	24 (27.9)	70 (27.7)	0.97	16 (27.1)	52 (28.9)	0.79	8 (29.6)	18 (24.7)	0.61
Yes	62 (72.1)	183 (72.3)		43 (72.9)	128 (71.1)		19 (70.4)	55 (75.3)	
Practice									
No	65 (75.6)	237 (93.7)	<0.01	48 (81.4)	170 (94.4)	<0.01	17 (63.0)	67 (91.8)	(<0.01)
Yes	21 (24.4)	16 (6.3)		11 (18.6)	10 (5.6)		10 (37.0)	6 (8.2)	
KAP pertaining to health-related applications									
Knowledge									
No	70 (81.4)	207 (81.8)	0.93	49 (83.1)	148 (82.2)	0.88	21 (77.8)	59 (80.8)	0.74
Yes	16 (18.6)	46 (18.2)		10 (16.9)	32 (17.8)		6 (22.2)	14 (19.2)	
Attitude									
No	37 (43.0)	75 (29.6)	0.02	27 (45.8)	60 (33.3)	0.09	10 (37.0)	15 (20.5)	0.09
Yes	49 (57.0)	178 (70.4)		32 (54.2)	120 (66.7)		17 (63.0)	58 (79.5)	
Practice									
No	81 (94.2)	234 (92.5)	0.60	55 (93.2)	164 (91.1)	(0.61)	26 (96.3)	70 (95.9)	(0.93)
Yes	5 (5.8)	19 (7.5)		4 (6.8)	16 (8.9)		1 (3.7)	3 (4.1)	

* *p*-values in parentheses were results of Fisher’s exact test.

**Table 4 ijerph-18-12704-t004:** Differences in KAP between two residential groups: multivariate logistic regression analysis ^a^.

Group Category	aOR ^b^	95% CI ^c^	*p* Value
Radiation-related devices and applications			
Knowledge			
The non-affected group ^d^	1.00 (Ref)		
The affected group ^d^	1.95	1.12–3.38	0.02
Practice			
The non-affected group ^d^	1.00 (Ref)		
The affected group ^d^	5.71	2.55–12.8	<0.01
Health-related applications			
Attitude			
The non-affected group ^d^	1.00 (Ref)		
The affected group ^d^	0.50	0.29–0.86	0.01

^a^ KAP (pertaining to radiation-related devices and applications and health-related applications) as the dependent variable, with characteristic information (gender, age range, education level) and knowledge of and concern about radiation (knowledge of radiation and concern about the risks of NPSs) as the adjusting variables. ^b^ Adjusted odds ratio. ^c^ Confidence interval. ^d^ Subject numbers of the non-affected and affected groups were 253 and 86, respectively.

**Table 5 ijerph-18-12704-t005:** Factors associated with attitude in the functioning of radiation- and the health-related applications using logistic regression analysis ^a^.

	aOR ^b^	95% CI ^c^	*p* Value	aOR ^b^	95% CI ^c^	*p* Value
Radiation-related devices and applications	Preferred Radi 1 ^d^			Preferred Radi 3 ^d^		
The non-affected group: N = 183	1.00 (Ref)			1.00 (Ref)		
The affected group: N = 62	0.50	0.26–0.95	0.04	0.43	0.23–0.81	0.01
Health-related applications	Preferred Health 4 ^d^			Preferred Health 5 ^d^		
The non-affected group: N = 178	1.00 (Ref)			1.00 (Ref)		
The affected group: N = 49	0.48	0.24–0.96	0.04	0.50	0.25–0.99	0.046

^a^ Preference (yes/sometimes) of each item as the dependent variable, and characteristic information (gender, age range, education level) and knowledge of and concern about radiation (knowledge of radiation and concern about the risks of NPSs) as adjusting variables. ^b^ Adjusted odds ratio. ^c^ Confidence interval. ^d^ Preferences for application functions: Radi (1) Measure environmental radiation level, Radi (3) Provide real-time information on the current situation (official channels only), Health (4) Provide general information on the effect of radiation on health and protection measures, and Health (5) Provide specific advice and instructions related to your personal situation and health status and/or well-being.

## Data Availability

Relevant questionnaire data have been presented herein. Other data are securely protected, but may be made available to qualified researchers upon reasonable request and in accord with local policy, national law, and the World Medical Association Declaration of Helsinki.

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
