# Peer review of "Tailoring Digital Tools to Address the Radiation and Health Information Needs of Returnees after a Nuclear Accident"

_ijerph, 2021, doi:10.3390/ijerph182312704_

Round 1

Reviewer 1 Report

An interesting paper.

Table 2:

Presence or absence of knowledge of radiation question

  • This is confusing.  Presence or absence of knowledge of radiation covers 100% of all possibilities.  A yes or no answer to this question is impossible

Author Response

Response to Reviewer 1 Comments

Thank you for valuable comments pertaining to ijerph-1452255, which prompted revisions as described below.

Point 1: An interesting paper.

Response 1: Thank you. Responses pertaining to your advice, and our revisions, are as follows.

Point 2: Table 2:

Presence or absence of knowledge of radiation question. This is confusing. Presence or absence of knowledge of radiation covers 100% of all possibilities. A yes or no answer to this question is impossible.

Response 2: Thank you for this important point, with which we concur. Regarding our binary “presence or absence of knowledge of radiation,” we re-analysed questionnaire data using self-evaluated levels of radiation knowledge in Table 2. These levels were categorised by four options: limited, average, professional, or none. Lines 204-210 address our revised analytical method for knowledge of radiation. In addition, we changed the multivariable analyses using logistic regression in Tables 4 and 5 to the knowledge of radiation with four options and re-analysed them. The richer data set did not overturn any results or conclusions. The values in our abstract and results section reflect the revised values in Tables 4 and 5.

Reviewer 2 Report

The content of this manuscript is interesting to readers, and the manuscript was also well organized. Although the survey sbujects are not sufficient, it is still worthy of publishing. Some minor revision are suggested: 1) The description of Line 119 is not well matched with the Figure 1, the NPSs can hardly be seen in the Figure. 2) It is recommended to change the subtile of 3.2 into "KAP of the two-age groups". 3)The form of Table 4 needs to be improved. 4) Asterisk (*) does not show neither in Fig. 2 nor Fig.3. 5)The sentence concerning about COVID-19 from Line 415 to Line 417 is suggested to be deleted as it is not closely related with the nuclear accident.  

Author Response

Response to Reviewer 2 Comments

Thank you for valuable comments pertaining to ijerph-1452255, which prompted revisions as described below.

Point 1: The content of this manuscript is interesting to readers, and the manuscript was also well organized. Although the survey subjects are not sufficient, it is still worthy of publishing. Some minor revision are suggested: 1) The description of Line 119 is not well matched with the Figure 1, the NPSs can hardly be seen in the Figure. 2) It is recommended to change the subtitle of 3.2 into "KAP of the two-age groups". 3)The form of Table 4 needs to be improved. 4) Asterisk (*) does not show neither in Fig. 2 nor Fig.3. 5)The sentence concerning about COVID-19 from Line 415 to Line 417 is suggested to be deleted as it is not closely related with the nuclear accident.

Response 1: Thank you very much for these important points, which prompted the following changes: 1) From line 119, we added and edited text to match Figure 1. To make it easier to see the position of each NPS, we enlarged the figure itself and also the dots representing each NPS. 2) We changed from “3.2 KAP of the two groups” to “3.2 KAP of the two-age groups” as suggested. 3) Indeed, Table 4 was complex. To avoid confusion, we listed Table 4 one item at a time. In addition, we add the information to our footnote in Table 4. 4) These were very important points for us. We added asterisks (*) to Figures 2 and 3. 5) We deleted text related to COVID-19 from line 454 to line 455.
